Comparison of iRoot BP Plus and mineral trioxide aggregate for  pulpotomy in primary molars under general anesthesia: a 3-year retrospective study

Zhao Yiming 1
Tao Yuyan 1
Wang Yan 2
Zou Jing 2
Zhang Qiong zhangqiongdentist@126.com 3
1 State Key Laboratory of Oral Diseases & National Center for Stomatology & National Clinical Research Center for Oral Diseases, West China Hospital of Stomatology, Sichuan University , Chengdu , Sichuan , China
2 State Key Laboratory of Oral Diseases & National Center for Stomatology & National Clinical Research Center for Oral Diseases, Department of Pediatric Dentistry, West China Hospital of Stomatology, Sichuan University , Chengdu , Sichuan , China
3 State Key Laboratory of Oral Diseases & National Center for Stomatology & National Clinical Research Center for Oral Diseases, Department of Pediatric Dentistry, Department of Jinjiang Outpatient, West China Hospital of Stomatology, Sichuan University , Chengdu , Sichuan , China
Abu Hasna Amjad
Electronic publication date: 2024 Nov 12
Publication date: 2024
Volume: 12
Electronic Location ID: e18453
Received 2024 Jun 11; Accepted 2024 Oct 14
Copyright: ©2024 Zhao et al.
Copyright year: 2024
Copyright holder: Zhao et al.
License: This is an open access article distributed under the terms of the Creative Commons Attribution License, which permits unrestricted use, distribution, reproduction and adaptation in any medium and for any purpose provided that it is properly attributed. For attribution, the original author(s), title, publication source (PeerJ) and either DOI or URL of the article must be cited.
License URL: https://creativecommons.org/licenses/by/4.0/

Keywords: Pulpotomy, Primary molars, MTA, iRoot BP Plus, Efficacy

Funding: Sichuan Science and Technology Program 2023NSFSC1509 2022NSFSC1482 This research was supported by the Sichuan Science and Technology Program (2023NSFSC1509 to Qiong Zhang and 2022NSFSC1482 to Yan Wang). The funders had no role in study design, data collection and analysis, decision to publish, or preparation of the manuscript.

==============================
Background

Pulpotomy is a widely recommended treatment for deep caries and reversible pulpitis in primary teeth. However, there is a significant lack of large-scale clinical studies evaluating the long-term efficacy of pulpotomy in primary molars, especially in studies with follow-up periods extending beyond a two years.

Aim

This study aimed to evaluate the long-term efficacy of mineral trioxide aggregate (MTA) and iRoot BP Plus for pulpotomy in primary molars performed under general anesthesia and to investigate factors influencing the success rate.

Methods

In this retrospective study, a total of 942 primary molars from 422 children who met the inclusion criteria underwent pulpotomy. Propensity score matching method (PSM) was used to match the MTA and iRoot BP Plus groups in a 1:1 ratio based on covariates. Efficacy was assessed using the Zurn & Seale criteria. Kaplan-Meier survival analysis and Cox proportional hazards model were performed to analyze the outcomes.

Results

PSM resulted in 266 pairs of matched teeth from 532 teeth of 291 children (mean age: 4.64 ± 1.07 years, ranging from 2 to 8 years). Long-term clinical and radiographic evaluations revealed higher success rates for iRoot BP Plus (24-month: 99.54%/97.25%; 36-month: 97.22%/95.83%) compared to MTA (24-month: 94.76%/95.29%; 36-month: 92.50%/91.25%). Survival analysis indicated a statistically significant difference between two groups (P = 0.0042). Age, gender, tooth position, and decayed tooth surface showed no significant impact on pulpotomy success, whereas the choice of pulp capping materials significantly influenced the outcome (HR [95% CI]=0.3745[0.1857-0.7552], P = 0.006).

Conclusion

Clinical and radiographic evaluations support the use of iRoot BP Plus for pulpotomy in primary molars.

Introduction

Pulpitis in primary teeth, predominantly caused by dental caries, represents a considerable challenge in pediatric dentistry (Kazeminia et al., 2020; Stringhini Junior et al., 2019). The cornerstone of managing primary molar caries lies in mitigating pulp inflammation and preserving pulp vitality (Coll et al., 2017). The main goals of clinical interventions include eliminating infection, alleviating pain, extending the lifespan of affected teeth, ensuring their functionality until physiological exfoliation, and preventing adverse effects on the developing permanent dentition (Duncan et al., 2019).

Pulpotomy is a critical procedure for preserving pulp vitality and is widely regarded as the preferred treatment for deep caries or reversible pulpitis in primary teeth (Bjørndal et al., 2019). The success of pulpotomy in primary molars depends on various factors, including proper case selection, accurate assessment of pulp infection extent, the nature of pulp exposure, the choice of pulp capping materials, and the efficacy of the coronal seal (Guo, Zhang & Cheng, 2023). The invasion by pathogenic microorganisms is a primary cause of pulp infection and a critical factor in pulpotomy failure, underscoring the importance of carefully selecting appropriate pulp capping agents (Bjørndal et al., 2019). For children with severe early childhood caries, general anesthesia enables the completion of all dental treatments in a single session, thus shortening the treatment period and reducing the caries risk in a short time frame. Research indicates that, for children with dental anxiety, comprehensive treatment under general anesthesia improves cooperation and compliance at follow-up visits, while also decreasing the incidence of secondary caries and the rate of filling failures (Liu et al., 2021).

In clinical practice, a diverse array of materials is available for pulpotomy. Traditional materials such as formocresol and calcium hydroxide are well-known for their antibacterial properties. However, concerns have been raised regarding the toxicity of formocresol (Olatosi, Sote & Orenuga, 2015) and the high pH of calcium hydroxide (Stringhini Junior, Vitcel & Oliveira, 2015), which may lead to pulp inflammation and internal/external root resorption. Recent advancements in dental material researches have introduced mineral trioxide aggregate (MTA) and bioceramic material iRoot BP Plus. These materials, both composed of calcium silicate, demonstrate advantageous biocompatibility, promoting the proliferation and differentiation of dental pulp cells (Emara, Elhennawy & Schwendicke, 2018; Zhang, Yang & Fan, 2013). Calcium silicate cement (CSC) are categorized into five generations, each characterized by advancements in composition and function. The first four generations of CSC are MTA, modified MTA, bioceramic materials, and hybrid cements (Dutta & Saunders, 2014), with the fifth generation representing the newly proposed high-performance bioceramics or smart biomaterials. Multiple studies have highlighted the potential benefits of using bioceramic materials in primary tooth pulpotomy. For instance, a two-arm meta-analysis revealed no significant difference (p > 0.05) between the use of MTA and non-MTA bioceramic-based materials as pulpotomy medicaments (Lin et al., 2024). Additionally, the one year success rate of pulpotomy using bioceramic putty was reported to be 95% in primary molars (Abdelwahab et al., 2024). A randomized controlled trial comparing MTA and bioceramic putty in primary molar pulpotomy with symptoms of irreversible pulpitis reported clinical and radiological success rates of 95% and 100%, respectively, after one year of follow-up (Alnassar et al., 2023). These findings suggest that bioceramic materials are a viable option for pulpotomy in primary molars.

MTA, approved by the U.S. Food and Drug Administration (FDA) (Parirokh, Torabinejad & Dummer, 2018), is a highly versatile material used for dental pulp treatment. ProRoot Gray MTA is a representative of the first generation of CSC (Dutta & Saunders, 2014), with calcium silicate, alumina, calcium oxide and silicon oxide as the main ingredients. Attributable to its analogous inorganic composition to dental hard tissue, MTA is capable of directly covering the pulp, thereby forming a dense dentin bridge that integrates seamlessly with the existing dentin (Nosrat, Seifi & Asgary, 2013). Additionally, MTA exhibits specific antibacterial properties and is non-toxic to surrounding tissues (Silva et al., 2019). However, despite these advantages, MTA encounters limitations in clinical application, including prolonged setting and operation times (Chang et al., 2014), mixing heterogeneity, potential for tooth discoloration (Marciano, Duarte & Camilleri, 2015; Silva et al., 2019), and complex application processes that may increase the risk of contamination (Lin et al., 2014). Recent advances in pulp treatment materials underscore the continued relevance of MTA, making it a suitable benchmark for evaluating the performance of newer materials such as iRoot BP Plus (Sanz et al., 2020; Wang et al., 2023).

iRoot BP Plus, a novel and premixed paste form of bioceramic material (Hu et al., 2023b) classified as the third generation of CSC, has demonstrated promising results in both in vitro studies and animal research, highlighting its biocompatibility (Zeng et al., 2023). The composition of iRoot BP Plus includes calcium silicate, zirconium oxide, tantalum pentoxide, anhydrous calcium sulfate, monobasic calcium phosphate, and filler agents. As a pulp treatment material, iRoot BP Plus exhibits physicochemical and biological properties similar to those of MTA, including stability, biocompatibility, antibacterial effects, sealing capabilities, and the potential to promote remineralization (Shi et al., 2016; Zhou et al., 2017). Recent studies have explored the varied applications of iRoot BP Plus, including apical sealing, pulp perforations repair, and the preservation of vital pulp tissue (Liu et al., 2020). However, most of these researches have focused on human dental pulp stem cells (Wang, Fangteng & Liu, 2019) and young permanent teeth (Yang et al., 2020), with primary teeth often overlooked. There is a noticeable scarcity of studies on the use of iRoot BP Plus in primary molar pulpotomy performed under general anesthesia.

In this study, a retrospective design was chosen due to its ability to utilize existing data for evaluating long-term outcomes, allowing for a broader patient sample. This retrospective study aimed to compare the clinical and radiographic outcomes of primary molars treated with either iRoot BP Plus or MTA in pulpotomy procedures conducted under general anesthesia. Furthermore, it intended to evaluate the long-term efficacy of pulpotomy in primary molars and to investigate the factors that contribute to the success rate of the procedure. The hypothesis was that iRoot BP Plus would be more effective than MTA in terms of long-term outcomes of pulpotomy in primary molars under general anesthesia, making it a more clinically recommended pulp capping material for pulpotomy in primary molars.

Materials & Methods

Subject selection

This retrospective study analyzed electronic medical records from the West China Hospital of Stomatology, Sichuan University. It included children diagnosed with deep caries or reversible pulpitis in primary molars and subsequently underwent pulpotomy under general anesthesia in the Department of Pediatric Dentistry from January 2019 to January 2021. Eligible teeth were selected based on predefined inclusion criteria. As a retrospective study, the selection of treatment materials was determined jointly by the patients and their dentists based on clinical judgment and personal preference, without randomization. To minimize bias from non-random grouping, propensity score matching (PSM) was employed in the data analysis to control for confounding factors and improve result reliability. All teeth treated with the pulp capping agent iRoot BP Plus were assigned to the experimental group (iRoot BP Plus group), and 1:1 matching using PSM was applied to include teeth treated with the pulp capping agent MTA in the control group (MTA Group). Basic patient information and relevant medical records were collected. Follow-up data collection was systematically planned to extend until December 2023, enabling a comprehensive evaluation of the long-term outcomes associated with the pulpotomy procedures. The data collection was completed in February 2024. In this retrospective study, the inclusion criteria were established based on available medical records.

Inclusion criteria:

• Primary molars diagnosed with deep caries or reversible pulpitis in patients aged less than 9 years.

• Absence of spontaneous pain, percussion pain, tooth mobility, or abnormal soft tissue symptoms.

• Preoperative radiographic examinations confirming no periapical abnormalities or pathology below the root bifurcation, no internal or external root resorption, and no canal obstruction.

• Patients unable to cooperate with outpatient treatment due to fear, classified as Class I by the American Society of Anesthesiologists Physical Status, with no prior history of general anesthesia.

• Pulpotomy treatment performed under general anesthesia, employing either iRoot BP Plus or MTA as the pulp capping material.

• A consistent follow-up regime that encompasses detailed documentation of clinical and radiographic evaluations.

Exclusion criteria:

• Patients with systemic diseases.

• Patients who received treatment for the affected tooth at an external hospital during the follow-up period.

This study was approved by the Ethic Committee of West China Hospital of Stomatology, Sichuan University (WCHSIRB-CT-2024-034). Written informed consent was obtained from the children’s legal guardians before treatment, explicitly agreeing to the use of their medical records for academic research.

Clinical procedures

The pulpotomy procedure in this study was carried out under a combination of intravenous and inhalation general anesthesia, involving a series of carefully executed steps to ensure effectiveness and safety. The primary investigator did not perform the treatment of the selected cases. The process, outlined in Fig. 1, was performed by experienced pediatric dental specialists, all of whom were also researchers involved in this study.

Figure 1 Clinical procedure of pulpotomy for the left mandibular second primary molar.

(A) Preoperative intraoral image displaying carious cavity extending to deep dentin. (B) Deep cavitated lesions radiographically extending into the pulpal quarter of dentin. (C) Rubber dam isolation following local anesthesia. (D) Opening of the carious cavity, removal of decay of lateral wall. (E) Removal of the decay of pulpal and axial wall, pulp perforation of the mesio-lingual pulp horn. (F) Exposure of the pulp chamber, where the pulp appeared bright red in color. (G) Irrigation of the pulp stump and gentle application of physiological saline wet cotton balls to achieve hemostasis. (H) Adequate hemostasis of the pulp stump. (I) Application of iRoot BP plus pulp capping agents over the pulp stump. (J) Cavity filling with glass ionomer cement. (K) Restoration of the tooth with a stainless steel crown.

• A comprehensive dental cleaning was conducted to remove plaque and calculus, and the patient’s occlusion was evaluated to identify any malocclusion that could impact the treatment.

• Local infiltration anesthesia was administered using lidocaine hydrochloride injection (CSPC Pharmaceutical Group Co, China) without adrenaline, followed by rubber dam isolation of the affected teeth.

• A tungsten steel ball bur, selected according to the extent of decay, was used to efficiently open the carious cavity, allowing for the removal of the decayed tissue from the cavity walls and the soft decay near the pulp.

• After removing the decay close to the pulp, the bur was changed as necessary to access the pulp chamber. The contaminated coronal pulp was excised using a sterile, sharp excavator, ensuring complete removal of all coronal pulp while preserving the radicular pulp, with care taken to avoid excessive pressure or trauma.

• The exposed pulp was thoroughly rinsed with a 1.5% sodium hypochlorite solution, followed by physiological saline to disinfect the area and remove any remaining debris.

• If bleeding ceased within three minutes, iRoot BP Plus (Innovative Bioceramix Inc, Canada) or ProRoot Gray MTA (DENSPLY Tulsa Dental Specialties, Tulsa, OK, USA) was applied over the pulp with a minimum thickness of two mm. The exposed tooth wall revealed the enamel-dentin junction, with the upper reference limit for the pulp-capping agent set two mm below this junction on the occlusal surface to ensure complete filling of the pulp chamber. A saline-soaked cotton ball was gently pressed onto the pulp capping agent to ensure it adhered well to the healthy pulp beneath.

• After the pulp capping agent was applied, the cavity base was filled with glass ionomer cement (GC Corporation, Tokyo, Japan) to provide a stable foundation for the final restoration.

• The tooth was then prepared and restored with a stainless steel crown (3M, Saint Paul, MN, USA) to ensure post-treatment durability and functional integrity.

Data collection

Data including patient sociobiological characteristics, examination records, and follow-up details were systematically collected. Sociobiological characteristics included the patient’s name, gender, date of birth, and treatment date. Examination details covered tooth position, extent of tooth surface decay, attending dentist, and imaging observations such as the depth of decay in dentin layers and potential decrease in bone density. Follow-up data comprised the follow-up date, primary complaints, status of crown restoration, clinical presentations, and radiographic findings. Crown restoration status addressed issues such as filling body fractures, secondary or recurrent caries, and detachment of stainless steel crowns. Clinical presentations consisted of functionality, normal or premature tooth loss, tooth mobility, gum health, and presence or absence of spontaneous or percussion pain. Radiographic findings assessed root bifurcation or periapical shadows, periodontal ligament width, root resorption, pulp canal obliteration, and dentin bridge formation.

Efficacy analysis

The efficacy evaluation utilized the Zurn & Seale primary dental pulp treatment clinical and radiographic examination scoring criteria (Table 1), with follow-up visits scheduled every 6 months post-treatment. According to the criteria, scores of 1 and 2 indicated success, while scores of 3 and 4 signified failure. For teeth with multiple radiographic and clinical examinations, the most severe score was recorded to ensure an accurate assessment of treatment outcomes. Teeth that failed during the post-treatment follow-up period and were subsequently re-treated or extracted were excluded from further follow-up analysis.

Table 1 Zurn & Seale primary dental pulp treatment clinical and radiographic examination scoring criteria.

Clinical score	Clinical symptom	Definition	
1	Asymptomatic	Pathology: Absent
Normal functioning
Natural replacement
Mobility (physiological) ≤1 mm	
2	Slight discomfort, short-lived	Pathology:Questionable
Percussion sensitivity
Gingival inflammation (due to poor oral hygiene)
Mobility (physiological)>1 mm, but <2 mm	
3	Minor discomfort, short-lived	Pathology: Initial changes present
Gingval swelling (not due to poor oral hygiene)
Mobility >2 mm, but <3 mm	
4	Major discomfort, long-lived extract immediately	Pathology: Late changes present
Spontaneous pain
Gingival swelling (not due to poor oral hygiene)
Periodontal pocket formation (exudate)
Sinus tract present
Mobility ≥3mm
Premature tooth loss, due to pathology	
Radiographic score	Radiographic finding	Definition	
1	No changes present at 6 months follow-up	Internal root canal form tapering from chamber to the apex
PDL/periapical regions: Normal width and trabeculation	
2	Pathological changes of questionable clinical significance at 3 months follow-up	External changes are not allowed (widened PDL) widening, abnormal
Inter-radicular trabeculation or variation in radiodensity
Internal resorption acceptable (not perforated)
Calcific metamorphosis is acceptable and defined as:Uniformly root canal; shape (non tapering); variation in radiodensity from canal to canal (one cloudier than the other)	
3	Pathological changes present at 1-month follow-up	External changes are present,but not large
Mildly widened PDL
Minor inter-radicular radiolucency with trabeculation still present
Minor external root resorption; internal resorption changes are acceptable,but not if external change is also present (perforated form)	
4	Pathological changes present extract immediately	Frank osseous radiolucency present	
Notes.

PDL Periodontal ligament

The calibration process involved assessing both intra-examiner and inter-examiner reproducibility. Two investigators independently completed a training and calibration program. Their findings were cross-referenced with those of an experienced pediatric dentist holding a senior title, and any diagnostic discrepancies were resolved by consensus. Inter-examiner reliability was evaluated, resulting in a kappa statistic measure of agreement of 0.879. Regarding intra-examiner reliability, one investigator reviewed the radiographs twice on separate occasions (κ = 0.865).

Statistical analysis

GPower 3.1 software was utilized to calculate the required sample size. Using t tests with the statistical test “Difference between two dependent means (matched pairs)”, an effect size of 0.3, an α = 0.05, and a power of 0.95, the total required sample size was 122, necessitating a minimum of 61 samples per group. This calculation is appropriate for determining sample sizes for groups matched through PSM.

SPSS 26.0 statistical software was employed for data analysis. The Shapiro–Wilk test was used to evaluate if the quantitative measurements followed a normal distribution. The Mann–Whitney U test, chi-square tests, and rank sum tests were utilized to assess significance based on variable characteristics. A significance level of α = 0.05 was established, with P<0.05 indicating statistical significance.

PSM was conducted using SPSS 26.0. The iRoot BP Plus group served as the benchmark, employing the logistic regression algorithm with adjusted variables as covariates. A caliper value of 0.02 was applied, and a 1:1 nearest neighbor matching method was utilized to match primary molars from the MTA group. The covariates considered for PSM included gender, age, maxillary vs. mandibular teeth, first primary molar vs. second primary molar, number of carious surfaces, presence of occlusal caries, and presence of axial caries.

For survival analysis and Cox proportional hazards model analysis, R software version 4.3.1 (R Core Team, 2023) was utilized. Kaplan–Meier survival analysis was performed to estimate survival times for the two groups, depicted through cumulative survival curves. Log-rank tests and overall comparison tests were applied to assess the treatment success across groups. The Cox proportional hazards model was used to analyze factors affecting success rates, with P<0.05 considered statistically significant.

Results

Baseline characteristics

A total of 942 primary molars from 422 children who met the inclusion criteria underwent pulpotomy. Among them, 645 primary molars from 269 children received MTA pulpotomy, while 297 primary molars from 153 children treated with iRoot BP Plus pulpotomy. Due to baseline inconsistencies and significant differences in case numbers between the two groups, PSM was employed to enhance comparability and mitigate selection bias. Gender, age, tooth position, decayed tooth surface, and follow-up duration were utilized as covariates for 1:1 matching, resulting in 266 pairs of matched teeth. The baseline characteristics of affected teeth before and after PSM is summarized in Table 2.

Table 2 Tooth information of MTA group and iRoot BP Plus group before and after PSM.

P-values are shown in bold.

Variable	Before PSM	P	After PSM	P	
		MTA (n = 645)	iRoot BP Plus (n = 297)		MTA (n = 266)	iRoot BP Plus (n = 266)		
Gender, n (%)								
	Male	340 (52.7)	172 (57.9)		156 (58.6)	149 (56.0)		
	Female	305 (47.3)	125 (42.1)	0.137	110 (41.4)	117 (44.0)	0.539	
lAge, years								
	Range	1.96∼8.59	2.35∼7.89		2.15∼8.07	2.53∼7.89		
	Median ± SD	4.59 ± 1.08	4.60 ± 1.01	0.623	4.62 ± 1.13	4.65 ± 1.00	0.435	
Tooth Position, n (%)								
	Maxillary	292 (45.3)	150 (50.5)		132 (49.6)	136 (51.1)		
	Mandibular	353 (54.7)	147 (49.5)	0.135	134 (50.4)	130 (48.9)	0.729	
	First primary molar	383 (59.4)	160 (53.9)		137 (51.5)	151 (56.8)		
	Second primary molar	262 (40.6)	137 (46.1)	0.112	129 (48.5)	115 (43.2)	0.223	
Number of decayed tooth surfaces, n (%)								
	1	34 (5.3)	51 (17.2)		34 (12.8)	27 (10.2)		
	2	145 (22.5)	131 (44.1)		111 (41.7)	124 (46.6)		
	3	180 (27.9)	75 (25.3)		74 (27.8)	75 (28.2)		
	4	122 (18.9)	23 (7.7)		29 (10.9)	23 (8.6)		
	5	164 (25.4)	17 (5.7)	0.000	18 (6.8)	17 (6.4)	0.690	
Occlusal caries, n (%)								
	Yes	8 (1.2)	8 (2.7)		6 (2.3)	6 (2.3)		
	No	637 (98.8)	289 (97.3)	0.109	260 (97.7)	260 (97.7)	1.000	
Axial caries, n (%)								
	Yes	34 (5.3)	51 (17.2)		34 (12.8)	27 (10.2)		
	No	611 (94.7)	246 (82.8)	0.000	232 (87.2)	239 (89.8)	0.341	
Follow-up								
	Range	6∼42	18∼42		6∼42	18∼42		
	Median ± SD	32.74 ± 8.61	28.18 ± 7.22	0.000	27.16 ± 8.50	28.26 ± 7.29	0.094	

Descriptive results of follow-up periods

Patients were scheduled for check-ups every 6 months post-pulpotomy. A portion of the participants were lost to follow-up due to various personal reasons. Furthermore, some treated teeth underwent normal physiological root resorption, which were anticipated and marked the end of the follow-up period for those specific teeth. Teeth reaching this stage were considered successfully treated, as the primary objective of pulpotomy is to preserve asymptomatic and functional primary teeth until their physiological exfoliation at the appropriate age. The follow-up progress of the two groups is illustrated in Fig. 2.

Figure 2 Flowchart depicting the follow-up status of included teeth within a 3-year period.

.

Descriptive results of clinical and radiographic outcomes

The efficacy evaluation was conducted using the Zurn & Seale scoring criteria for clinical and radiographic examinations post-treatment. Tables 3 and 4 present the clinical and radiographic scores obtained throughout the follow-up period. iRoot BP Plus demonstrated a higher success rate in both clinical and radiographic outcomes compared to MTA. Normal physiological tooth loss was considered a measure of success. In the MTA group, five out of 266 teeth (1.9%) underwent physiological replacement during the follow-up, while in the iRoot BP Plus group, 19 out of 266 teeth (7.1%) experienced physiological replacement.

Table 3 Clinical scores during follow-up.

Follow-up	MTA (n = 266)	iRoot BP Plus (n = 266)	
	Clinical scores	Clinical success rate	Clinical scores	Clinical success rate	
	1	2	3	4		1	2	3	4		
6 months	265	0	1	0	99.62%	266	0	0	0	100.00%	
12 months	260	0	1	1	99.24%	266	0	0	0	100.00%	
18 months	248	1	3	3	97.65%	265	0	0	1	99.62%	
24 months	180	1	5	5	94.76%	217	0	0	1	99.54%	
30 months	119	2	3	4	94.53%	138	0	2	1	97.87%	
36 months	73	1	2	4	92.50%	70	0	0	2	97.22%	

Table 4 Radiographic examinations scores during follow-up.

Follow-up	MTA (n = 266)	iRoot BP Plus (n = 266)	
	Radiographic scores	Radiographic success rate	Radiographic scores	Radiographic success rate	
	1	2	3	4		1	2	3	4		
6 months	263	2	0	1	99.62%	266	0	0	0	100.00%	
12 months	257	2	3	0	98.85%	264	0	0	2	99.25%	
18 months	245	2	5	3	96.86%	262	0	0	4	98.50%	
24 months	179	3	4	5	95.29%	212	0	1	5	97.25%	
30 months	118	0	4	6	92.19%	137	0	1	3	97.16%	
36 months	73	0	3	4	91.25%	69	0	1	2	95.83%	

The main reasons for clinical failure were chronic apical periodontitis (MTA: 21 cases; iRoot BP Plus: four cases) and chronic pulpitis (iRoot BP Plus: one case). Radiographic failures were primarily due to internal root resorption (MTA: three cases; iRoot BP Plus: one case), external root resorption (MTA: 12 cases; iRoot BP Plus: six cases), and furcation involvement (MTA: 11 cases; iRoot BP Plus: two cases). Pulp canal obliteration was regarded as a healing reaction of pulp tissue and an acceptable outcome for pulp treatment. Figure 3 illustrates typical X-ray radiographic features indicative of such outcomes. Table 5 details the distribution of reasons for radiographic failure at the 24-month and 36-month follow-up.

Figure 3 Illustrates various radiographic features observed in the study.

(A) Normal tooth radiograph. (B) Dentin bridge in distal root canal. (C) Distal pulp canal obliteration caused by root canal calcification (D) Internal root resorption. (E, F) External root resorption. (E) Minor external root resorption. (F) Severe external root resorption. (G–I) Furcation involvement. (G) Minor furcation radiolucency or variation in radiodensity. (H) Furcation radiolucency with preserved trabeculation. (I) Furcation radiolucency with distal root resorption and involvement of the dental follicle of the permanent successor.

Table 5 Radiographic features at 24 months and 36 months follow-up.

Radiographic features (24 months)	MTA (n = 191)	iRoot BP Plus (n = 218)	
Pulp canal obliteration, n (%)	10 (5.24)	15 (6.88)	
Internal root resorption, n (%)	2 (1.05)	1 (0.46)	
External root resorption, n (%)	5 (2.62)	4 (1.83)	
Furcation involvement, n (%)	3 (1.57)	1 (0.46)	
Radiographic features (36 months)	MTA (n = 80)	iRoot BP Plus (n = 72)	
Pulp canal obliteration, n (%)	6 (7.50)	7 (9.72)	
Internal root resorption, n (%)	–	–	
External root resorption, n (%)	6 (7.50)	3 (4.16)	
Furcation involvement, n (%)	2 (2.50)	2 (2.78)	

Survival analysis and comparison of two types of pulp capping agents

Kaplan–Meier survival analysis comparing the long-term efficacy of the two pulp capping materials is presented in Fig. 4. The log rank test results revealed a statistically significant difference between the two materials (P = 0.0042), indicating that iRoot BP Plus exhibited superior long-term efficacy compared to MTA.

Figure 4 Kaplan-Meier survival time curve based on the final results of maintaining asymptomatic teeth after pulpotomy using two types of pulp capping materials.

.

Regression analysis of factors influencing surgical success

The Cox proportional hazards model analysis identified significant associations between the survival time of teeth treated with pulpotomy and the type of pulp capping material used. As shown in Table 6, the type of pulp capping material emerged as a significant predictor of surgical success. Conversely, neither gender nor age significantly impacted the success rate. The success rate remained consistent across different dentists.

Table 6 Cox proportional hazards model analysis of factors influencing success rate.

Variable	HR [95% CI]	Std. error	Z	P >(Z)	
Gender					
	Male (ref)					
	Female	0.7297[0.3773-1.411]	0.3365	−0.937	0.349	
Age		1.2257[0.9279-1.619]	0.1420	1.433	0.152	
Tooth position						
	Maxillary (ref)					
	Mandibular	1.6462[0.8587-3.156]	0.3321	1.501	0.133	
	First primary molar (ref)					
	Second primary molar	1.2318[0.6513-2.33]	0.3251	0.641	0.521	
Decayed tooth surface					
	Occlusal caries	0.4979[0.1196-2.073]	0.7278	−0.958	0.338	
	Axial caries	1.2229[0.4336-3.449]	0.5291	0.38	0.704	
Number of decayed tooth surfaces	1.0275[0.749-1.41]	0.16128	0.168	0.866	
Dentist	1.1330[0.9586-1.339]	0.08526	1.464	0.143	
Material					
	MTA (ref)					
	iRoot BP Plus	0.3745[0.1857-0.7552]	0.3579	−2.744	0.006*	
Notes.

* P < 0.05

Concerning tooth position, although the risk of treatment failure was higher in mandibular teeth compared to maxillary teeth, and higher in the second primary molar compared to the first primary molar, these differences lacked statistical significance. Additionally, neither the type of decay (occlusal caries vs. axial caries) nor the number of decayed tooth surfaces significantly impacted the surgical success rate.

This analysis highlights the critical role of the type of pulp capping material in the success of pulpotomy treatments, while other factors such as patient demographics, tooth position, and extent of decay did not exhibit a significant effect.

Discussion

Pulpotomy is a viable treatment option for deep carious pulp exposure and reversible pulpitis in both primary teeth and young permanent teeth. The aim of this retrospective study was to assess and compare the long-term efficacy of iRoot BP Plus and MTA in pulpotomy of primary molars, while also examining factors influencing the success rate. This study hypothesized that iRoot BP Plus would exhibit superior effectiveness compared to MTA in pulpotomy procedures for primary molars performed under general anesthesia. The results supported this hypothesis, with iRoot BP Plus showing higher clinical success rates and fewer post-operative complications than MTA.

In this retrospective study, PSM method was employed to match individuals from the two groups based on selected confounding factors, aiming to achieve balance between the groups and minimize selection bias. PSM is primarily employed to reduce bias and the effects of confounding variables in observational studies, thereby clarifying treatment effects in the experimental group (Rosenbaum & Rubin, 1983). The PSM approach begins with logistic regression to estimate the probability of each individual receiving a specific treatment or intervention based on covariates, producing a propensity score. After calculating propensity scores, the caliper matching method is employed to match the most similar samples from the MTA group with those from the experimental group (iRoot BP Plus group). In this study, the caliper value was set at 0.02, meaning that only individuals with a propensity score difference within 0.02 were matched. After matching, descriptive statistics and balance tests (such as standardized mean differences) were conducted to assess the balance of baseline characteristics between the two groups, resulting in experimental and control groups that were directly compared. In addition to the application of PSM, this study utilized a sample size exceeding that required for the effect size, which helps to reduce the influence of confounding factors related to inter-operator procedural variation and subject selection.

The efficacy of clinical and radiographic outcomes following pulp treatment in primary teeth was evaluated using the Zurn & Seale scoring criteria. Unlike binary outcome measures of “success” or “failure”, this scoring system offers a graded assessment of post-treatment conditions, reflecting the severity of lesions observed during follow-up. Higher scores indicate more severe outcomes, providing a nuanced understanding of the healing process and the effectiveness of the treatment. The Zurn & Seale criteria encompass clinical indicators (such as pain, tenderness to percussion, abscess, swelling, fistula, and pathological mobility) as well as radiographic findings (including radicular radiolucency, internal and external root resorption, periodontal ligament space widening, and furcation radiolucency). This scoring system has been widely used in studies of primary tooth pulpotomy, demonstrating reliability in pediatric populations (Abdelwahab et al., 2024; Faghihi et al., 2021; Sharaf et al., 2023; Subramanyam & Somasundaram, 2020).

In this study, iRoot BP Plus outperformed MTA in both clinical and imaging success rates, showcasing enhanced stability in maintaining dental pulp health and consistent calcification barrier formation. The long-term follow-up revealed a statistically significant survival advantage for iRoot BP Plus (P = 0.0042). Previous meta-analyses and randomized clinical trials have reported the 2-year efficacy of MTA in primary tooth pulpotomy to be between 90% to 100% (Ansari & Ranjpour, 2010; Stringhini Junior et al., 2019), consistent with our findings. A retrospective analysis, with an 18-month median follow-up, reported iRoot BP Plus achieving clinical and radiographic success rates of 98.9% and 95.5%, respectively, which closely align with our results (Hu et al., 2023b). Additionally, a randomized clinical trial indicated that the success rates for iRoot BP Plus in primary tooth pulpotomy were 96%, 92%, and 87% at 3, 6, and 12 months, respectively (Wang et al., 2021). The superior 2-year success rate observed in this study could be attributed to the standardized treatment procedures under general anesthesia, which has been shown to reduce the incidence of secondary caries, new caries, and re-treatment compared to outpatient clinics (Liu et al., 2021).

In cases where pulpotomy treatment failed, chronic apical periodontitis and irreversible pulpitis were the primary clinical diagnoses during follow-up. Additionally, there may be external root resorption, internal root resorption, and furcation involvement observed in imaging. The MTA group exhibited a higher incidence of adverse reactions compared to the iRoot BP Plus group, indicating that iRoot BP Plus leads to better postoperative outcomes. The formation of pulp canal obliteration can increase the sealing of the root canal system and improve the efficacy of pulpotomy (Careddu & Duncan, 2021). This study observed pulp canal obliteration in both treatment groups, with rates of 5.24% and 7.50% in the iRoot BP Plus group and 6.88% and 9.72% in the MTA group at 24 and 36 months post-pulpotomy, respectively.

The superior therapeutic efficacy of iRoot BP Plus over MTA is likely attributed to its physicochemical composition and biological characteristics.

iRoot BP Plus features a finer texture, making it more suitable to inflammatory acidic environments, and it demonstrates reduced cytotoxicity compared to MTA. One of the key distinctions between the two materials is the smaller particle size and homogeneous nanoparticle components of iRoot BP Plus, which effectively stimulate mineral deposition and promote osteogenesis (Zeng et al., 2023). Furthermore, iRoot BP Plus can locally generate an alkaline environment through the release of silicon ions and calcium ions, which exerts antibacterial effects. This attribute enhances its effectiveness in acidic inflammatory pulp conditions (Tian et al., 2017). Additionally, the substitution of bismuth oxide with tantalum oxide in iRoot BP Plus reduces cytotoxicity and minimizes adverse reactions, further improving its clinical utility (Hu et al., 2023a).

In terms of biological characteristics, both in vitro and in vivo studies have demonstrated that iRoot BP Plus surpasses MTA in enhancing adhesion, migration, mineralization, and differentiation of primary dental pulp stem cells into bone. Confocal immunofluorescence staining has revealed that iRoot BP Plus stimulated focal adhesion formation and stress fiber assembly in dental pulp cells, while also upregulating the expression of focal adhesion molecules (Zhu et al., 2014). Additionally, alkaline phosphatase, a key marker for new bone formation, showed higher activity in stem cells from human exfoliated deciduous teeth and dental pulp stem cells treated with iRoot BP Plus compared to those treated with MTA (Zhang, Yang & Fan, 2013). Studies have further revealed that iRoot BP Plus significantly outperforms MTA in promoting cell mineralization (Wang, Fangteng & Liu, 2019). Quantitative reverse transcriptase analysis indicated that dental pulp stem cells treated with iRoot BP Plus expressed higher levels of mineralization-related genes and formed apatite crystals more effectively (Zhang, Yang & Fan, 2013).

The factors influencing the success rate of pulpotomy have always been controversial. It is commonly believed that younger patients typically exhibit enhanced pulp regenerative potential, leading to more favorable treatment prognoses (Massler, 1972). This age-related variation could theoretically impact the healing capacity of damaged dental pulp, thereby influencing treatment success rates. However, contemporary clinical trials and systematic reviews, particularly those focused on permanent teeth, have consistently demonstrated that pulpotomy is an effective treatment option across all age groups (Kang et al., 2017). In this study, patients ranged from 2 to 8 years old, and no statistically significant difference in the success rate of tooth treatment was observed among different age groups. This finding aligns with recent clinical research, further confirming the efficacy of pulpotomy regardless of age.

Gender has also been demonstrated not to influence the success of pulpotomy treatment. A randomized clinical trial specifically focusing on mature permanent teeth with clinical signs of irreversible pulpitis found that gender did not significantly impact the clinical and radiographic outcomes of partial pulpotomy (Taha & Khazali, 2017). This conclusion is supported by another randomized clinical trial (Kang et al., 2017), further reinforcing the idea that gender is not a determining factor in pulpotomy success.

In this study, neither the position of the teeth nor the surface of the decay significantly affected the success rate of pulpotomy. This finding is consistent with previous research, including a five-year follow-up study of pulpotomy in permanent teeth, which also concluded that tooth type (anterior teeth, premolars, molars) and the position of pulp exposure (occlusal cavity, adjacent cavity, adjacent occlusal cavity) were not related factors affecting postoperative success (Tan et al., 2020). While some studies on permanent teeth suggest that adjacent exposure may lead to poorer outcomes due to challenges in caries removal, cavities sealing, and application of pulp capping materials (Liu et al., 2020), this study did not observe this scenario. Furthermore, it was revealed that the number of decayed tooth surfaces did not significantly affect the success rate. This lack of significance may be attributed to the fact that all treated teeth were restored with stainless steel crowns after pulpotomy, potentially minimizing the differences between teeth with varying numbers of carious tooth surfaces. The uniformity in restoration likely contributed to the consistent success rates observed, regardless of the extent of decay.

Admittedly, this study had some limitations and the results should be interpreted with caution. The sample size decreased with increasing follow-up time. Although the sample size with a follow-up period of 3 years meets the minimum requirements to obtain statistically powerful results, large-sample randomized controlled clinical trials are anticipated to verify the findings.

In this study, different dentists followed a standardized treatment protocol, including the use of a rubber dam for moisture isolation, the same pulp cavity irrigation solution, uniform thickness of the pulp capping agent, and restoration with stainless steel crowns. Coronal microleakage is recognized as a significant factor contributing to prolonged pulpotomy failure, with the coronal sealing material directly influencing its efficacy (Yazdani et al., 2014). Previous similar studies have also used stainless steel crowns as final restorations due to their superior coronal sealing ability and durability (Alnassar et al., 2022). However, the retrospective nature of this study, where stainless steel crowns were consistently utilized for coronal restoration, limits the exploration of the impact of varied coronal restoration methods on pulpotomy success rates. This constitutes a limitation of this study.

Successful pulpotomy in clinical practice requires additional assessments, including evaluation of intraoperative coronal pulp texture, bleeding characteristics, and post-pulpotomy hemostasis time (Dhar et al., 2017; Hu et al., 2023a). Cases involving pulp abscess, pulp necrosis, or persistent bleeding may necessitate consideration of root canal therapy as an alternative intervention (Hu et al., 2023b). Future research may benefit from incorporating a broader range of operational variables and larger sample sizes to establish a more comprehensive reference basis for clinical practice.

Conclusions

iRoot BP Plus pulpotomy is an effective treatment for deep caries or reversible pulpitis in primary molars, demonstrating superior outcomes compared to MTA in reducing adverse events. The success of the treatment is consistent across different gender, age, affected tooth position, and the number of decayed tooth surfaces, making iRoot BP Plus a reliable choice for clinical practice.

Supplemental Information

Data S1 Raw data before PSM

The information of all patients recorded before propensity score matching, and the matching process.

Data S2 Raw data after PSM

The patient information after propensity score matching, which minimizes selection bias and enhances comparability between groups. Subsequent efficacy evaluations of the two materials are performed based on this refined dataset.

Additional Information and Declarations

Competing Interests

Author Contributions

Human Ethics

Ethics

Data Availability

The authors declare there are no competing interests.

Yiming Zhao conceived and designed the experiments, performed the experiments, analyzed the data, prepared figures and/or tables, authored or reviewed drafts of the article, and approved the final draft.

Yuyan Tao performed the experiments, analyzed the data, prepared figures and/or tables, authored or reviewed drafts of the article, and approved the final draft.

Yan Wang performed the experiments, authored or reviewed drafts of the article, and approved the final draft.

Jing Zou conceived and designed the experiments, authored or reviewed drafts of the article, and approved the final draft.

Qiong Zhang conceived and designed the experiments, prepared figures and/or tables, authored or reviewed drafts of the article, and approved the final draft.

The following information was supplied relating to ethical approvals (i.e., approving body and any reference numbers):

This study was conducted with approval from the Ethic Committee of West China Hospital of Stomatology, Sichuan University (WCHSIRB-CT-2024-034). All the procedures complied with the principles of the Helsinki Declaration.

The following information was supplied relating to ethical approvals (i.e., approving body and any reference numbers):

This study was conducted with approval from the Ethic Committee of West China Hospital of Stomatology, Sichuan University (WCHSIRB-CT-2024-034). All the procedures complied with the principles of the Helsinki Declaration.

The following information was supplied regarding data availability:

Raw data are available in the Supplemental Files.

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
