# Peer review of "Comparison of iRoot BP Plus and mineral trioxide aggregate for pulpotomy in primary molars under general anesthesia: a 3-year retrospective study"

_PeerJ, doi:10.7717/peerj.18453_

## Round 0.1 · original submission · Major Revisions

Dear Authors,

I appreciate your time and effort in preparing this manuscript. However, I have a concern regarding the use of general anesthesia in your study. Additionally, please define the study's hypothesis in the introduction section and clearly state whether you accept or reject it in the discussion section.

Kindly address all the reviewers' comments and questions thoroughly to avoid further rounds of peer review.

Thank you.

Reviewer 1 ·

Basic reporting

The manuscript is well-written in good English and uses a minimum number of references. However, the article could benefit from more citations. The images may be adequate but need to be more precise. The hypotheses related to this study were not mentioned.

Experimental design

The submitted work is original and falls within the scope of the journal. The research question might not be explicitly stated, and the topic of pulpotomy in primary teeth using calcium silicate cements in general and bioceramics in particular is not new. The procedures followed are somewhat methodologically sound, but the description of the pulpotomy process needs to be more detailed to ensure reproducibility.

Validity of the findings

I don't believe the topic addressed possesses novelty, as it has been previously explored. The data was handled well in terms of organization and statistical analysis. The results are relevant to the work presented.

Additional comments

The mechanism employed for PSM needs to be explained comprehensively to understand how some patients were excluded without bias.
I believe the term "Atresia" is not commonly used in dentistry, and a more precise term should be adopted.
Additionally, it would be clearer to use "bur" rather than "drill".
Specific tests for the distribution of children's gender and ages across the groups should be conducted similarly to those in this article: [https://pubmed.ncbi.nlm.nih.gov/38485988/].

Reviewer 2 ·

Basic reporting

The language used in the current study needs linguistic revision by a native English-speaking dentist (some terms are not used in dentistry, such as: Chronic revisable pulpitis and Atresia). References supporting the content were used, but some are unrelated to the topic or may have been used by mistake. I have objections to the provided images, which I will address later. The researchers did not present any hypotheses at the end of the introduction section to be discussed in the discussion section.

Experimental design

The topic presented in the current design is contemporary, although the research question is not entirely clear. It is necessary to cite more related studies that have used bioceramic materials in pulpotomy of primary teeth. I have concerns regarding the use of general anesthesia. The authors did not mention any reasons for treating children under general anesthesia. Were they children with special needs, did they have dental anxiety, or were they uncooperative?

Validity of the findings

The details mentioned in the current study may not be sufficient for replicating the study again. For example, how was the thickness of the applied material calibrated? Was the local anesthetic applied with adrenaline? What is the manufacturer of the local anesthetic used? By the way, most manufacturer names are not provided. The level of coronal pulp cutting is unclear. The statistical analyses used are appropriate, although tests for normal data distribution, for example, were not provided. The conclusions are relevant.

Additional comments

Both the references Tewari and Rao were used in an irrelevant manner and should be replaced with appropriate references (in this case, all references need to be re-checked).
There is a difference between the materials used for direct pulp capping and those used in pulpotomy. Please do not confuse them in the introduction and discussion.
Calcium silicate cements are classified into five generations and differ in the composition of powders and liquids, whether they are pre-mixed or not. What generation of MTA is used? What is its manufacturer? Is it white or gray? All this should be added in the materials and methods section, and the discussion about these generations and the components of the materials used should be expanded in the introduction section.
You can use these studies as references for discussion and citation: https://pubmed.ncbi.nlm.nih.gov/36579231/, https://pubmed.ncbi.nlm.nih.gov/36464977/
Please do not use the first person in scientific writing; use the passive voice instead.
Why were children who experienced dental trauma excluded? Most trauma affects the front teeth, not the back teeth. There should be a rationale or citation regarding this matter.
Coming to the issue of age distribution of the study participants, the age range proposed by the researchers is too large, and tests showing the age distribution of patients between the two groups should be provided.
This age distribution plays a significant role in children's pain perception (and thus the reliability of clinical criteria). Moreover, the authors mentioned that all children underwent general anesthesia! Is this because the patients were fearful? Dental fear affects the reliability of clinical outcomes and is a confounding factor that influences the overall results.
The thickness of the material used and the level of pulp cutting are confounding factors that could affect the study results. In the presented radiographic images of the study cases, I found that the authors were not consistent with either of the previous factors. There were cases of direct pulp capping while retaining the pulp horns and cases of pulpotomy up to the canal openings with complete removal of the chamber roof. This varies significantly in terms of procedure, as the area of pulp exposure to the restorative material is crucial as a potential success or failure factor.
Finally, the evaluation criterion used (Zurn and Seale) needs a citation. Is it reliable for children?

Reviewer 3 ·

Basic reporting

1. Language are clear and concise.
Some technical aspects on writing references. Can refer my comment on annotated PDF.

2. Literature and background are well written and just enough to direct the readers to understand the topic.

3. All tables are well displayed.

Experimental design

Aim was clear and method were well planned.

My only concern on dentists/operator as this factor may have effect on treatment outcomes. Were all the operators dental specialists?

Validity of the findings

No comment. well written.

Annotated reviews are not available for download in order to protect the identity of reviewers who chose to remain anonymous.

---

## Round 0.2 · Minor Revisions

The manuscript's validity has improved, but further minor revisions are necessary before it can be considered for publication.

Reviewer 2 ·

Basic reporting

I still believe that the submitted manuscript is weak linguistically and requires correction using a Manuscript Editing Service.
The images remain unconvincing; it is very clear from images in Figure 3b, h, and i that the level of pulp amputation is not uniform across all cases.
The introduction, after incorporating the required revisions, has become disjointed.

Experimental design

he two main issues in the materials and methods are as follows:

Firstly, the large age range of the children may render the scale reliable for older age groups but unreliable for younger age groups, which is a significant problem.

Secondly, according to the researchers' clarification, the children included are those who suffer from dental fear. Will this group of children provide reliable results for post-treatment pain values?

Validity of the findings

The manuscript has been improved by adding some details related to the materials and methods to make it reproducible.

Additional comments

MTA ProRoot is a first-generation calcium silicate cement, not a second-generation one what is the generation of I Root BP ?
plaque and tartar ?? tartar ???!
The reference "Mahgoub et al., 2019" is not related
The reference style indicated that you should mentioned the first authors' second name not the first. Check all the reference accordingly.
The term "chronic reversible pulpitis " that you used in the manuscript is not scientifically corrected. it should be stated like "reversible pulpitis" only.
Revise this sentence "In clinical practice, a diverse array of pulp capping materials is available for pulpotomy" into "In clinical practice, a diverse array of materials is available for pulpotomy"
You mentioned in your rebuttal message "The exclusion of children who experienced dental trauma was based on the consideration that dental trauma in very young children might affect their cooperation during subsequent follow-up visits. This potential impact on compliance was a key factor in their exclusion from the study. ". Please cite this sentence and add it into the related exclusion criteria or delete this sentence!

Reviewer 3 ·

Basic reporting

Language used are clear. Corrections made as per suggested. Hypothesis was mentioned.

Experimental design

Research gap is stated which includes lack of study on the use of iRoot BP Plus in primary molar pulpotomy performed under general anesthesia.

The authors should also highlight reasons of the procedures were performed under general anaesthesia during decision making. As this will act as cofounding variables too.

Validity of the findings

The results which are reported, corresponding to the objective and method.

Conclusions should be simplified. Kindly highlighted the 'take home messages' to the reader and not repeating things in discussion.

---

## Round 0.3 · Minor Revisions

Dear authors,

The manuscript would benefit from revisions for clarity and completeness. The abstract should include the age criteria of participants and avoid repetitive details about testing methods. In the introduction, further justification for using gray MTA and the retrospective study design is needed. The methodology should clarify that the primary investigator did not perform the treatments, and provide more information on the age criteria, given the long-term follow-up. Additionally, uniformity in referencing patterns is essential for consistency throughout the manuscript.

Reviewer 4 ·

Basic reporting

Highlights:

• The abstract provides a concise summary of the research, delivering a clear overview of the key findings.
• The introduction effectively establishes the context and justification for the study, emphasizing the need for evaluating long-term efficacy of mineral trioxide aggregate (MTA) and iRoot BP Plus for pulpotomy in primary molars performed under general anesthesia
• Detailed information is provided on the study design, and the retrospective study methodology and evaluation parameters

Recommended Corrections:
• Abstract: Clarity on Age criteria of the patients included in the study for a long term follow up as it is a retrospective study, eliminate repetitions of testing methods, and enhance clarity on the study design. The abstract provides a concise overview of the research, effectively summarizing the findings. However, to enhance its clarity and effectiveness, the age criteria of included cases needs to be added as the study is assessing the long term effect of pulp capping materials.

• Introduction: The need of selection of retrospective study design can be mentioned in the introduction. Selection of gray MTA to compare it with iRoot BP Plus needs a justification.
The introduction successfully highlights the generations of Calcium Silicate Cement as pulpotomy material. To further enrich the discussion, it is advisable to justify selection of gray MTA to compare it with iRoot BP Plus, based on various research studies. This not only broadens the context but also positions the current study within the recent advances in pulpotomy materials. Furthermore, explicitly stating reason for selection of retrospective study design will enhance the justification of the study.

Experimental design

2. Experimental Design:

Highlights:

• Utilization of propensity score matching (PSM).
• comprehensive evaluation of the long-term outcomes associated with the pulpotomy procedures.
• Inclusion and exclusion criteria are mentioned for selection of cases from records, ensuring study integrity.
• efficacy evaluation utilising the Zurn & Seale primary dental pulp treatment clinical and radiographic examination scoring criteria

Recommended Corrections:

• In the inclusion criteria, since it’s a retrospective study controlling all the mentioned parameters during the case allotment to the selected materials is highly difficult.
The methodology section has mentioned elaborate inclusion criteria, but since it’s a retrospective study here the selection of cases will not be done by the primary investigator. Controlling all these parameters will not be possible until the original dentist who treated the cases has mentioned the detail description of the selected cases and the follow up visit assessments in the case records

• The primary investigator has not performed the treatment of the selected cases, so the detailed description of the procedure mentioned in the manuscript may not have been followed by the dentist who treated the cases.
When the treatment was rendered the present study design was not framed, so strict adherence to the case selection and clinical procedure may not have been followed. Since the study involved more than the sample size required for the effect size it will easily overcome the confounding factors of these procedural variations and selection issues. This information is essential for authenticity and a more thorough understanding of the experimental setup.


• Include information on the age of the participants as the study involved long term follow up so age of the included samples will be an essential parameter.
The study would benefit from including details regarding the cut of age for inclusion in the study as the primary Molars has limited survival before physiologic exfoliation. This information is pertinent to understanding the baseline information especially when a long term effect is assessed.

Validity of the findings

3. Validity of the Findings:

Highlights:

• PSM method to enhance comparability and mitigate selection bias
• Efficacy evaluation using the Zurn & Seale scoring criteria for clinical and radiographic examinations post-treatment.
• Kaplan-Meier survival analysis comparing the long-term efficacy of the two pulp capping materials
• The discussion elaborates on the results considerably, connecting them to existing literature and highlighting potential implications.

Recommended Corrections:

• In the discussion, avoid repeating statements about MTA. Focus more on result-oriented discussion.
While the discussion effectively highlights the study parameters and its positive results on pulpotomy, there is a tendency to repeat statements regarding MTA which was already mentioned in the introduction. To enhance the quality of the discussion, it is advisable to minimize repetitions and focus on a more result-oriented analysis. Delving deeper into the implications of the results, potential applications in dentistry, and any limitations encountered during the study will contribute to a more robust and insightful discussion.


• Provide a uniform reference pattern throughout the manuscript
In the discussion section line number 309, the reference authors names been repeated and do not follow the pattern followed in the other areas of the manuscript.

• Please check for spelling in the line number 381.
Check the following statement for spelling “The factors influencing the success rate of pulpotomy have always been controversial. It is commonly believed that younger individuals possess stronger repair abilities within their pulp tissue, leading to better surgical prognoses (Massler 1972).”

• Follow uniform reference patterns
The recommendations related to references include addressing formatting issues, such as following a consistent pattern related to the volume and page numbers of the articles.


Some of the noted errors

• Ibrahim A, Mohamed A, Mohammad Salem R, Hasan A, and Anas A. 2022a. Evaluation of the efficacy of mineral trioxide aggregate and bioceramic putty in primary molar pulpotomy with symptoms of irreversible pulpitis (a randomized-controlled trial). Clinical and Experimental Dental Research.

• Ibrahim A, Mohamed K A, Mohamad Salem R, Hasan A, and Imad K. 2022b. Evaluation of Bioceramic Putty in Pulpotomy of Immature Permanent Molars With Symptoms of Irreversible Pulpitis. Cureus 14. 10.7759/cureus.31806

Additional comments

4. General Comments:

The study is carried out with a good study design, requiring corrections in the mentioned points and uniform referencing patterns.
My comments highlight the overall novelty of the study and its commendable study design. However, to elevate the quality of the manuscript, it is crucial to address the specific points mentioned in my review. Achieving uniformity in referencing patterns will enhance the professionalism and consistency of the manuscript.

---

## Round 0.4 · accepted · Accept

Dear authors,

Thank you for your revised manuscript. After careful review, I am pleased to inform you that all the suggested changes have been appropriately incorporated, and the necessary corrections to the experimental design have been made. With no further concerns regarding the validity of the findings, and all feedback addressed, I am happy to recommend acceptance of your manuscript for publication.

Reviewer 4 ·

Basic reporting

All the suggested changes have been incorporated.

Experimental design

Authors have done the suggested corrections

Validity of the findings

No comment

Additional comments

The manuscript has been revised with suggested corrections.